# Crystal structure of dopamine D1 receptor in complex with G protein and a non-catechol agonist

Bingfa Sun [1,7], Dan Feng[1,7], Matthew Ling-Hon Chu[1], Inbar Fish[1], Silvia Lovera[2], Zara A. Sands[2,6], Sebastian Kelm [3], Anne Valade[2], Martyn Wood[2], Tom Ceska [3], Tong Sun Kobilka[1], Florence Lebon[4] & Brian K. Kobilka [1,5✉]

Dopamine D1 receptor (D1R) is an important drug target implicated in many psychiatric and neurological disorders. Selective agonism of D1R are sought to be the therapeutic strategy for these disorders. Most selective D1R agonists share a dopamine-like catechol moiety in their molecular structure, and their therapeutic potential is therefore limited by poor pharmacological properties in vivo. Recently, a class of non-catechol D1R selective agonists with a distinct scaffold and pharmacological properties were reported. Here, we report the crystal structure of D1R in complex with stimulatory G protein (Gs) and a non-catechol agonist Compound 1 at 3.8 Å resolution. The structure reveals the ligand bound to D1R in an extended conformation, spanning from the orthosteric site to extracellular loop 2 (ECL2). Structural analysis reveals that the unique features of D1R ligand binding pocket explains the remarkable selectivity of this scaffold for D1R over other aminergic receptors, and sheds light on the mechanism for D1R activation by the non-catechol agonist.

[1] ConfometRx, Inc., Santa Clara, CA, USA. [2] UCB Pharma, Braine-l'Alleud, Belgium. [3] UCB Pharma, Slough, UK. [4] UCB Pharma, Anderlecht, Belgium. [5] Department of Molecular and Cellular Physiology, Stanford University School of Medicine, Stanford, CA, USA. [6] Present address: Confo Therapeutics, Zwijnaarde, Belgium. [7] These authors contributed equally: Bingfa Sun, Dan Feng. ✉email: kobilka@stanford.edu

Dopamine is a critical catecholamine neurotransmitter in the central nervous system, responsible for the control of locomotion, cognition, emotion, reward, and regulation of the endocrine system[1,2]. Abnormalities in dopaminergic signaling are associated with a wide range of psychiatric and neurological disorders[1–5]. Dopamine binds to dopamine receptors on the plasma membrane and initiates complex signaling events downstream. Dopamine receptors are a family of G protein-coupled receptors (GPCRs) with five subtypes, which can be further divided into D1-like (D1R, D5R) and D2-like (D2R, D3R, and D4R) subfamilies based on their sequence similarities and G protein coupling specificity[1,2]. For decades, the dopamine D1 receptor (D1R) is thought to be the therapeutic target for Parkinson's disease, attention deficit hyperactivity disorder and cognitive impairment in schizophrenia, where normal D1R signaling is impaired[1–5]. Upon activation by dopamine, D1R undergoes conformational changes and couples to the intracellular heterotrimeric stimulatory G protein (Gs), which in turn activates adenyl cyclase and elevates the level of the second messenger cyclic AMP (cAMP) level to drive downstream signaling processes. In addition to this canonical G protein dependent pathway, many GPCRs, including D1R, also signal through non-canonical G protein independent pathways. After activation by dopamine, D1R recruits β-arrestin2 and is internalized by endocytosis, resulting in loss of receptor on the cell surface and additional arrestin-mediated signaling events[6–8].

Restoration of D1R signaling to optimal levels in desired pathways, without altering the function of other closely related receptors is critical for the treatment of the disorders mentioned above[9]. Over the years, many subtype selective D1R agonists were developed and tested as clinical candidates[10]. Although many of them are potent D1R agonists in biochemical and cellular assays, they did not behave well in vivo as a result of poor bioavailability and rapid metabolism, due to their shared catechol moiety[10–13]. Furthermore, prolonged exposure to these agonists results in tachyphylaxis in vivo due to desensitization caused by the activation of the arrestin pathway as mentioned above[6–8]. These drawbacks limit the potential of therapeutic application of these compounds.

It is hypothesized that non-catechol agonists and allosteric modulators can overcome these drawbacks[6,10,13–16]. Recently, a class of non-catechol agonists was reported with an interesting efficacy profile[6,17,18]. They are highly selective towards D1R and D5R, versus other dopamine receptor subtypes and closely related aminergic receptors[17]. They show potent partial agonist or full agonist efficacy on cAMP pathway, and many of them are much less effective in stimulating the recruitment of β-arrestin2 by the D1R[6,17]. As a result of their functional selectivity, they caused little or no internalization of D1R in cells and have more sustained behavioral activity in animal models[6]. These results highlighted the potential of non-catechol agonists when targeting D1R for therapeutic purposes.

To further the understanding of the activation mechanism of D1R by non-catechol agonists and facilitate drug discovery, we determined the crystal structure of D1R in complex with Gs and a non-catechol agonist, Compound 1 (reported in international patent WO2014072881). The structure reveals unique features of the Compound 1 binding pocket in D1R and provides insights into the subtype selectivity of this scaffold, and the mechanism of D1R activation in the absence of a catechol moiety.

## Results

**Overall structure of D1R in complex with Gs and a non-catechol agonist.** We have determined the crystal structure of D1R in complex with Gs and a the non-catechol ligand, (-)-6-[4-furo [3,2-c]pyridin-4-yloxy)-2-methylphenyl]-1,5-dimethylpyrimidine-2,4(1H,3H)-dione (Compound 1), using the lipidic cubic phase (LCP) crystallization method (Fig. 1a). The structure was determined at 3.8 Å resolution (Statistics in Supplementary Table 1) and most D1R residues could be confidently located and modeled according to density, including the majority of their side chain orientations, except for residues 242–261 in intracellular loop (ICL) 3, and residues 165–184 in extracellular loop (ECL) 2. For the trimeric G protein, Gβ, and Gγ subunits were also clearly seen in the density map, as well as the core Ras-like domain of Gαs subunit. The alpha-helical domain (AHD) of the Gαs subunit is less well defined and its position is different from what is observed in β2 adrenergic receptor (β2AR)-Gs structure (Supplementary Fig. 1a), consistent with previous studies showing that the AHD becomes flexible in a nucleotide free form, when a receptor-G protein complex is formed[19,20]. D1R is in a fully active conformation, featured by an open conformation of the cytoplasmic end of TM6 and active-state specific conformations of DRY and NPXXY motifs, which are conserved throughout family-A GPCRs (Fig. 1b and Supplementary Fig. 1b)[21,22]. In general, D1R adopts a very similar conformation to the active-state of β2AR, with an overall RMSD of 1.0 Å for the receptor Cα atoms between D1R reported here and the β2AR structure coupled to Gs[19].

One of the most prominent difference between the two receptor structures is the longer helical extension at the cytoplasmic end of

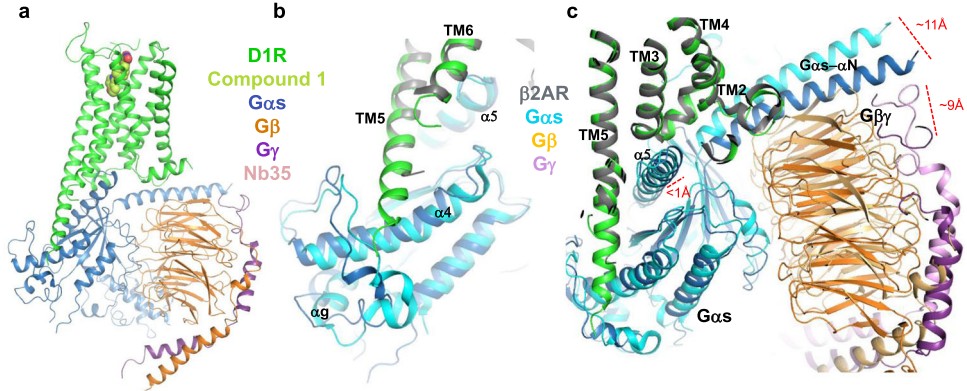

**Fig. 1 Overall structure of D1R in complex with Gs and non-catechol agonist Compound 1. a** The structure of D1R and Gs are shown in cartoon, and ligand Compound 1 is shown in spheres. D1R: green; Compound 1: lime; Gαs: blue; Gβ: orange; Gγ: purple; Nb35: light pink. **b** Superimposition of D1R-Gs structure with β2AR-Gs structure by receptor (gray for β2AR, cyan for Gαs, light orange for Gβ, violet for Gγ; PDB ID: 3SN6) shows a longer TM5-ICL3 helix in D1R, which interacts with the α4 helix and α4-αg loop of Gαs. **c** Differences in the relative orientation of the Gs trimer and receptor for D1R-Gs and β2AR-Gs structures following superimposition of the receptors.

TM5 observed in D1R compared to β2AR (Fig. 1b). Upon coupling to Gs, the TM5 helical structure in β2AR is extended by about seven residues[19], compared to uncoupled structures, while in D1R, it is extended by about 14 residues based on sequence alignment. The extended TM5-ICL3 helix in D1R is in close proximity to the α4 helix and the α4-αg loop of Gαs subunit and likely interacts with them. These interactions support a different conformation of these structural elements compared to β2AR-Gs structure, and as a consequence, a different orientation of the heterotrimer relative to the receptor (Fig. 1c). When superimposing the two receptors, the distances between the N-terminus of αN in the Gαs subunit of the D1R-Gs and β2AR-Gs complexes are approximately 11 Å apart, and the C-terminal residues in the Gβ subunit are approximately 9 Å apart, while the Cα of residues throughout α5-helix in Gαs are less than 1 Å apart. Similar variances in G protein heterotrimer orientation versus receptor were also observed in multiple cryo-EM structures of GPCR-Gs protein complexes obtained in detergent micelle environments (Supplementary Fig. 2)[23–27]. It was proposed that the differences arise from distinct interactions of receptors with Gs heterotrimer (reviewed in the ref. [28]), through ICL1 and ICL2, as in A2AR[26], or helix-8 (H8) as in family B GPCRs[23–25,27]. Since these regions of D1R are highly similar to β2AR, the difference we observed here are likely to be associated with the extended helical structure of TM5.

**Ligand binding mode.** The electron density for Compound **1** (Fig. 2a and Supplementary Fig. 3a) extends from the orthosteric site of D1R to ECL2. The modeled ligand fits the density well (Supplementary Fig. 3a) with real space correlation coefficient at 0.86, and reasonable B factor compared to the entire receptor (138 Å² for Compound **1** vs. 154 Å² for D1R). To be noted, due to limited resolution, and the extended "linear" backbone shape of the ligand, we cannot rule out the possibility that there are additional conformations to model the ligand, or that there is a mixture of conformations. We strongly favor our presented conformation due to the density indicating the position of methyl group on the middle phenyl ring (Supplementary Fig. 3a). Furthermore, four independent 1 μs molecular dynamics (MD) simulations of Compound **1** revealed that the ligand maintained its binding pose, having an average heavy atoms RMSD of 0.6 ± 0.2 Å (± indicate standard deviation, S.D.) with respect to the crystallographic pose (Fig. 2b and Supplementary Fig. 3b, c). Due to the ambiguity of the detailed ligand binding pose, we will limit our discussion and conclusions to ligand–receptor interactions, where we have the greatest confidence. A unique feature of Compound **1** and its analogs is the lack of the positively charged nitrogen atom that exists in almost all other dopamine receptor ligands, which interact with Asp103[3.32] (Ballesteros–Weinstein numbering[29]). In the structure, Asp103[3.32] is within interaction distance of Compound **1**, but no polar interaction can be identified. The mutation of Asp103[3.32], an absolutely conserved residue across all aminergic receptors, resulted in total loss-of-function for dopamine, while this mutant can still be activated by non-catechol agonists at reduced potency[6], indicating it is contributing to the interaction with Compound **1** but not as critical as with dopamine, consistent with the structure model.

Compound **1** binds to D1R in a fully extended conformation, with its uracil ring directly interacting with ECL2 (Fig. 2a, b). This results in a distinct conformation for residues 186–191, the part of ECL2 between TM5 and the conserved cysteine residue in our D1R structure. For all aminergic receptors with structures determined (including but not limited to those reported in the refs. [30–39]), the conformation of this loop allows interactions with ligands at a much deeper position, while in D1R, this loop shows a significant bulge to accommodate the rigid and extended

Compound **1** molecule (Fig. 2c). A main chain hydrogen bond is formed between the nitrogen of Compound **1**'s uracil ring and the Cys186 main chain carbonyl oxygen (Fig. 2b). It is noteworthy that ECL2 is quite flexible in MD simulation, and this hydrogen bond is not maintained (Supplementary Fig. 3b). Another important interaction contributed by ECL2 is through the side chain of Leu190, interacting with both the uracil and furo-pyridine rings at the ends of the extended Compound **1** molecule (Fig. 2b, d), with Leu190 alone contributing a large complex interface surface area of 73.7 Å². These observations are consistent with mutagenesis data regarding Compound **1**'s close analogs, suggesting that both Leu190 and Ser188 on ECL2 are important for their potency[6]. The mutation of Leu190 was shown to cause the most significant decrease in potency among all binding pocket residues tested[6]. Interestingly, the mutagenesis data suggested that Ser188 is not interacting with the ligand through its side chain, since a serine to alanine mutation did not affect any of the compounds tested, while a serine to isoleucine mutation reduced their potencies[6]. This is explained by our structure model and MD simulations, where it is observed that this part of ECL2 bulges outward with the side chain of Ser188 facing solvent (Fig. 2c), in which case mutating to a bulkier hydrophobic side chain would be unfavorable.

Other elements of Compound **1**, which consists of a central phenoxy ring, the connecting heteroatom oxygen, and furo-pyridine ring, are considered to be the core scaffold of this ligand, and are more sensitive to modifications. The central phenoxy ring of Compound **1** fits in a channel adjacent to the orthosteric pocket with more space on the side towards TM6, where the methyl group is located (Fig. 2d). It was shown that any substitution at a position other than the methyl group on Compound **1** greatly impaired ligand potency[17], which can be explained by this binding pose (Supplementary Fig. 4a). The furo-pyridine ring sits in the orthosteric site, where endogenous dopamine is predicted to bind[40]. It interacts with surrounding residues through extensive aromatic and hydrophobic interactions. It is also within interaction range of Ser107[3.36], Thr108[3.37], Ser198[5.42], and Ser202[5.46], but the distances are too long for significant hydrogen bonds to form, which is further supported by MD simulations (Supplementary Fig. 3b). Consistent with this observation, mutations of Ser198[5.42] and Ser202[5.46] have minor effects on the potency of Compound **1**'s close analogs, while their mutations cause major loss of potency for dopamine[6]. However, it is possible that some analogues with this scaffold may bind slightly deeper or may be tilted to form one or more hydrogen bonds with these residues. Together, all these residues form a relatively tight pocket around the furo-pyridine ring, with little space for additional substituents. Structure–activity relationship studies showed that a fused aromatic moiety at this position is required for strong potency in this scaffold[17], consistent with our structure model (Supplementary Fig. 4b).

**Structural basis for subtype selectivity.** The scaffold of this non-catechol agonist shows remarkable selectivity towards D1R/D5R over D2R-like dopamine receptors and other aminergic receptors[17]. The binding mode revealed by our structure provides insight into the mechanism behind it. First, our structure indicates the critical role of ECL2 in Compound **1** binding. Comparison of ECL2 sequence of D1R to other dopamine and aminergic receptors reveals features unique to D1R/D5R (Fig. 3a). Specifically, the +2 position after the conserved cysteine residue on ECL2 is Ser188 in D1R, which is distant from the orthosteric site with its side chain facing bulk solvent, while in all other dopamine, serotonin and adrenaline receptors, the respective position is always a larger hydrophobic residue (Fig. 3a). In all available structures for these

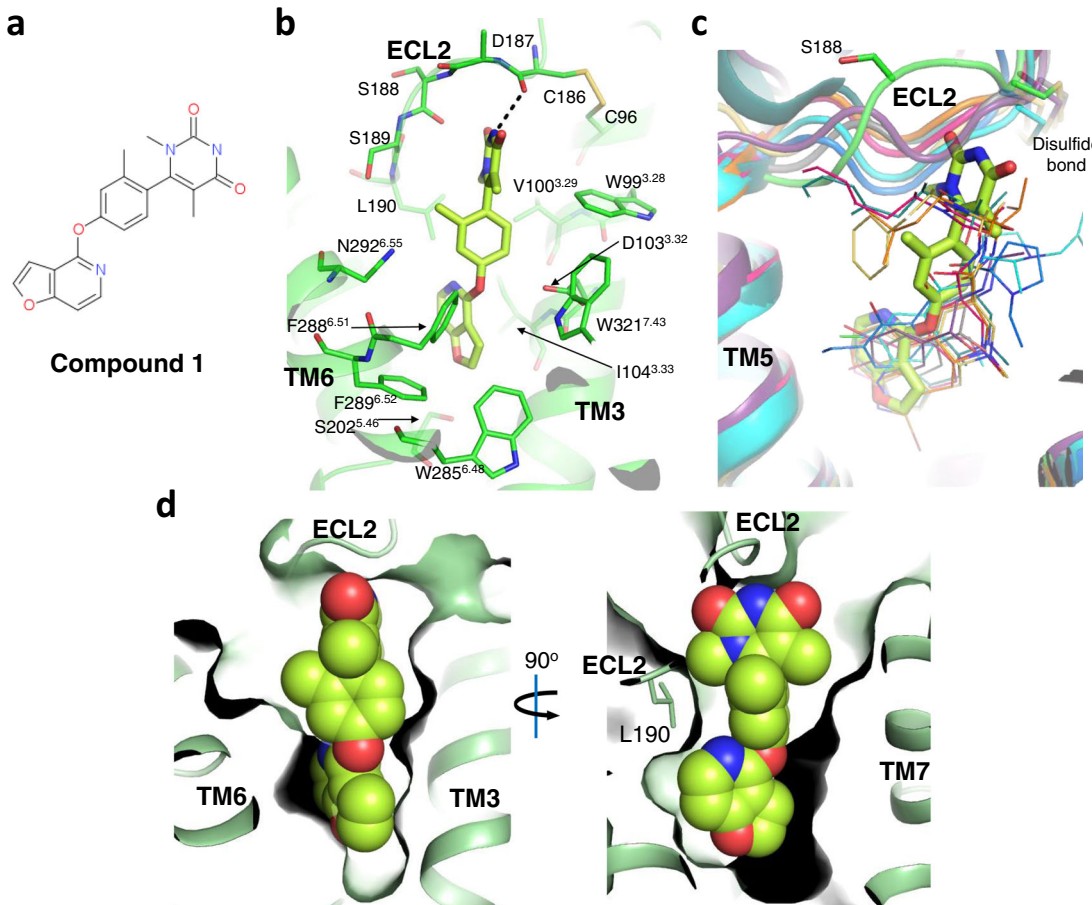

**Fig. 2 The binding of non-catechol agonist Compound 1 to D1R. a** The chemical structure of Compound **1**. **b** Binding pocket of Compound **1**. Residues within the interaction distance of the ligand are shown in stick. Hydrogen bonds are depicted with dashed lines. **c** ECL2 of D1R adopts a unique bulged conformation to accommodate Compound **1** (shown in stick). D1R structure is aligned with multiple aminergic receptor structures. All other bound ligands are shown in lines. D1R: green, Compound **1**: lime; β2AR, gray (PDB ID: 4LDO); D2R: orange (PDB ID: 6VMS); 5-HT2C, magenta (PDB ID: 6BQG); 5-HT1B, yellow (PDB ID: 4IAR); β1AR, purple (PDB ID: 2Y03); D4R, cyan (PDB ID: 5WIV); D3R, blue (PDB ID: 3PBL); 5-HT2B, dark green (PDB ID: 4IB4). **d** Surface representation of Compound **1** binding pocket. Compound **1** is shown in spheres, while the side chain of Leu190 on ECL2 of D1R is shown in stick.

closely related aminergic receptors (including but not limited to those reported in the refs. [30–38]), the side chain of this residue is always facing towards the core orthosteric site and stabilizing ligand binding (Fig. 3b). Another residue unique to D1R/D5R, Leu190 at the +4 position after the conserved cysteine residue on ECL2 (Fig. 3a, b) is also critical for the activity of compounds bearing the same scaffold as Compound **1**[6]. Most closely related aminergic receptors have a polar residue at this position (Fig. 3a).

Another noticeable feature of the Compound 1 pocket in D1R is the presence of Trp321[7.43] (Fig. 3c) at the orthosteric site deep in the receptor core. In other aminergic receptors, with the exception of the closely related D5R, there is a less bulky tyrosine residue at this position (Fig. 3a). The difference in this key residue most likely contributes to the unique pharmacological profile for D1R and D5R. For example, ergotamine is a promiscuous aminergic receptor ligand with nanomolar affinity to many of them, but only shows weak affinity for D1R and D5R[34]. Previous structural studies revealed the residues in other aminergic receptors making direct interactions with ergotamine, and they are fairly conserved in D1R and D5R, especially the residues involved in binding with its ergoline core. Superimposing our D1R structure with the structures of 5-HT1B[35], 5-HT2B[32] in complex with ergotamine reveals significant steric conflict of Trp321[7.43] in D1R with the methyl group attached to ergotamine's basic nitrogen, preventing its binding (Fig. 3c). Similar steric conflict is observed when aligning the structure of D2R bound to an ergoline

derivative bromocriptine (Fig. 3c). On the other hand, lysergic acid diethylamide (LSD), another ergoline derivative, shows similar affinity towards all subtypes of dopamine receptors, including D1R[41]. The structure of 5-HT2B bound to LSD[31] revealed that the ergoline moiety of LSD is located higher and more tilted compared to that of ergotamine in 5-HT2B and bromocriptine in D2R (Fig. 3c). As a result, when the structures are aligned, the methyl group attached to the basic nitrogen is predicted to have minimal steric conflict with Trp321[7.43] in D1R, consistent with its D1R pharmacology.

To summarize, the combination of Trp321[7.43] and ECL2 defines a unique pocket in D1R and its close homologue D5R for Compound **1**. These features are likely the key for explaining the remarkable selectivity of this scaffold of non-catechol agonists towards D1R/D5R over D2R-like dopamine receptors, as well as other aminergic receptors[17].

**Structural basis for receptor activation by non-catechol agonist.** Compound **1** is a partial agonist of D1R in the cAMP signaling pathway when tested in Lmtk cells (Fig. 4a), although it also behaved as full agonist in another cell background[42]. It was reported that compounds with this non-catechol scaffold exhibit a range of efficacy from partial to full agonism in the cAMP pathway[6,17], while the mechanism by which they activate the

**a**

| GPCR | ECL2 | | | | | 7.43 |
|------|------|------|------|------|------|------|
| | C | +1 | +2 | +3 | +4 | |
| **D1** | **C** | **D** | **S** | **S** | **L** | **W** |
| **D5** | **C** | **D** | **S** | **S** | **L** | **W** |
| D2 | C | I | I | A | N | Y |
| D3 | C | S | I | S | N | Y |
| D4 | C | R | L | E | D | Y |
| β1 | C | D | F | V | T | Y |
| β2 | C | D | F | F | T | Y |
| β3 | C | A | F | A | S | Y |
| α1A | C | Q | I | N | E | Y |
| α1B | C | G | V | T | E | Y |
| α1d | C | G | I | T | E | Y |
| α2a | C | E | I | N | D | Y |
| α2b | C | K | L | N | Q | Y |
| α2c | C | G | L | N | D | Y |
| 5-HT1A | C | T | I | S | K | Y |
| 5-HT1B | C | V | V | N | T | Y |
| 5-HT1D | C | L | V | N | T | Y |
| 5-HT1E | C | T | I | Q | H | Y |
| 5-HT1F | C | I | I | K | H | Y |
| 5-HT2A | C | L | L | A | D | Y |
| 5-HT2B | C | V | L | T | K | Y |
| 5-HT2C | C | V | L | N | D | Y |
| 5-HT4 | C | V | F | M | V | Y |
| 5-HT5 | C | Q | V | S | R | Y |
| 5-HT6 | C | R | L | L | A | Y |
| 5-HT7 | C | L | I | S | Q | Y |

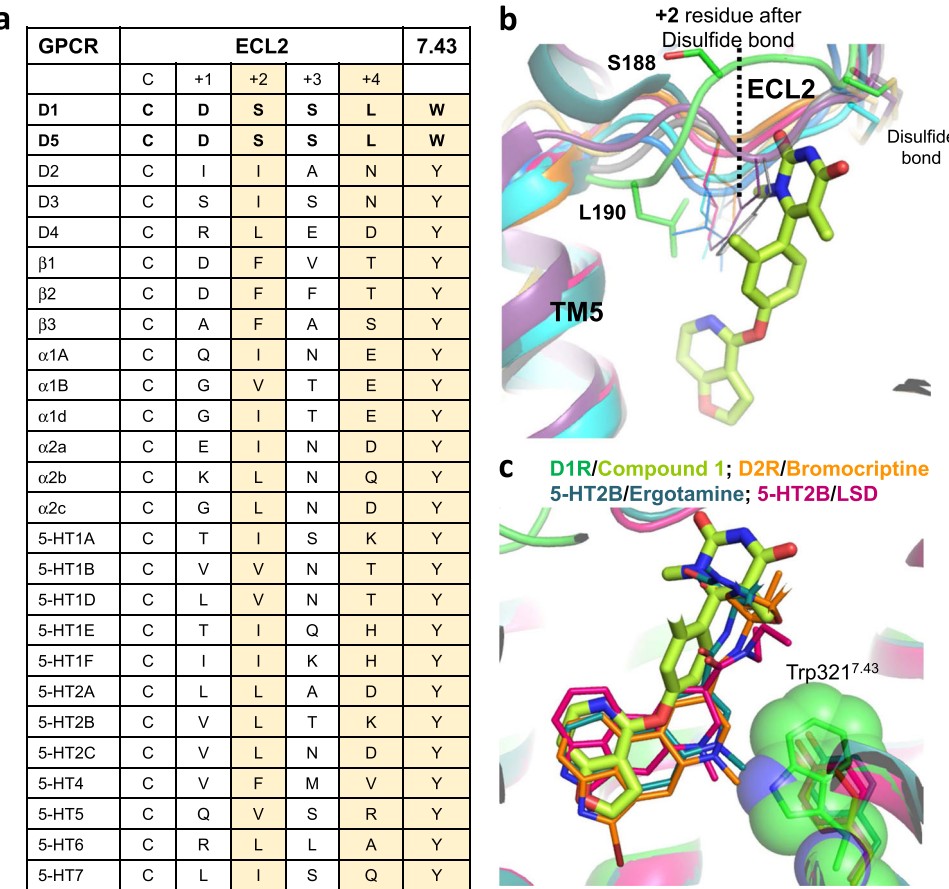

**Fig. 3 Unique features of the D1R ligand binding pocket. a** Sequence alignment of aminergic receptors at ECL2, starting from the conserved cysteine, as well as 7.43 position (Ballesteros–Weinstein numbering). The residues at cysteine+2, cysteine+4 and 7.43 positions are highlighted, as they are unique in D1R/D5R. **b** The side chain orientations of the +2 residues (position indicated by dashed line) on ECL2, counting from the conserved disulfide bond forming cysteine residue. Compound **1** and Ser188 (cysteine+2 residue), L190 (cysteine+4 residue) of D1R, as well as all disulfide bonds are shown in stick, while cysteine+2 residues on other aminergic receptors are shown in line. D1R: green, Compound **1**: lime; β2AR, gray (PDB ID: 4LDO); D2R: orange (PDB ID: 6VMS); 5-HT2C, magenta (PDB ID: 6BQG); 5-HT1B, yellow (PDB ID: 4IAR); b1AR, purple (PDB ID: 2Y03); D4R, cyan (PDB ID: 5WIV); D3R, blue (PDB ID: 3PBL); 5-HT2B, dark green (PDB ID: 4IB4). **c** Trp321[7.43] is a unique feature in the D1R orthosteric site. D1R structure is aligned with 5-HT2B bound to ergotamine (dark green, PDB ID: 4IB4) and LSD (hot pink, PDB ID: 5TVN), and D2R bound to bromocriptine (orange, PDB ID: 6VMS). The Van der Waals radius of Trp321[7.43] side chain is shown in sphere.

receptor without a catechol moiety is not clear. As revealed by the structure of the β2AR bound to adrenaline[30], catechol compounds form hydrogen bonds with Ser[5.42] and Ser[5.46] on TM5 of β2AR through the double hydroxyl groups of the catechol moiety. These two interactions result in an inward movement of Pro[5.50] and subsequently the rearrangement of PIF motif and opening of intracellular side of TM6, which is considered as a common mechanism for the activation of aminergic receptors[21,22]. When aligning the structure of D1R bound with the non-catechol agonist to β2AR, its closest homologue with structure reported, no inward movement of Pro206[5.50] is observed compared to the inactive-state of the β2AR, while Ile111[3.40] and Phe281[6.44] of the PIF motif are both in their active conformation (Fig. 4b and Supplementary Fig. 5b)[19,30,43]. The distances between Cα atoms of Ile3.40 and Pro5.50 in β2AR-Gs and D1R-Gs structure are 6.9 Å and 7.5 Å, respectively. More importantly, the closest distance of between atoms of Pro5.50 and Ile3.40 in β2AR-Gs is 3.8 Å, while in D1R-Gs this distance is 4.9 Å. Therefore, there is no direct interaction of Pro5.50 and Ile3.40 in the D1R-Gs structure. This indicates that the activation of D1R by Compound1 is not driven by its interaction with TM5, neither is it required for the active conformation of Ile3.40 and Phe6.44 in the PIF motif, in contrast to what is observed with catechol agonists.

Detailed analysis revealed that Compound **1**'s furo-pyridine ring interaction with side chains of Trp285[6.48], Phe288[6.51], and Phe289[6.52] on TM6 of D1R is likely to be the trigger for D1R activation. Comparing to inactive-state positions of conserved residues found in the inactive-state structure of β2AR, the extensive interactions among these four aromatic elements (Supplementary Fig. 5c) result in Trp285[6.48] being shifted to a deeper position (Fig. 4c), which is likely to be the driving force for the repositioning and rotation of Phe281[6.44] in the PIF motif (Fig. 4d). The side chain of Phe289[6.52] also shifts to a position deeper and closer to TM5 (Fig. 4c), from where it causes a shift of Phe203[5.47] side chain through strong aromatic stacking (Fig. 4d). The conformation of Phe203[5.47] creates space which facilitates the rearrangement of Ile111[3.40] and Phe281[6.44] in the PIF motif to the active conformation (Fig. 4d and Supplementary Fig. 5d). The network of aromatic/hydrophobic interactions between Compound 1 and surrounding aromatic residues, as well as interactions between all abovementioned residues, are maintained and stable throughout multiple independent MD simulations (Supplementary Fig. 5e). In support of this mechanism, previously reported mutagenesis studies showed that mutations of Phe288[6.51] and Phe289[6.52] significantly impaired the potencies of Compound **1**'s close analogs, while the mutations of Ser198[5.42] and Ser202[5.46] have minor effect on them[6]. Among the aromatic residues,

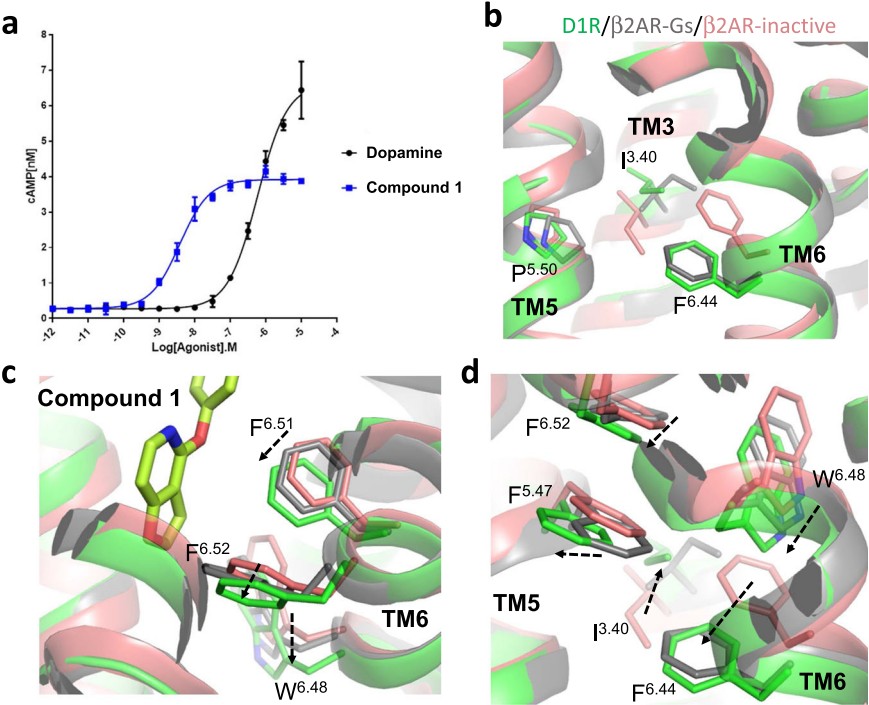

**Fig. 4 Structural basis for D1R activation by the non-catechol agonist Compound 1. a** Compound **1** is a partial agonist of D1R compared to dopamine when tested in Lmtk cells. Dose-dependent cAMP stimulation curves of Compound **1** (blue) and dopamine (black) are generated by GraphPad Prism5.0. The curves shown are typical from one experiment with triplicate determinations, repeated at least twice ($n = 2$). Points and error bars on each curve represent the mean ± SEM. Source data are provided as a Source Data file. **b** The PIF motif conformation of D1R-Gs (green), β2AR-Gs (gray, PDB ID: 3SN6), β2AR inactive-state (pink, PDB ID: 2RH1) structures. The side chains of residues are shown in stick. **c** Compound **1** (lime) binding to D1R induces a shift in the positions of Phe[6.51], Phe[6.52], Trp[6.48] through its aromatic furo-pyridine ring, relative to the inactive-state position of homologous residues in the β2AR. The dashed arrows indicate estimated residue shift during activation. **d** The shift of Trp[6.48] drives the movement of Phe[6.44] to the active conformation, while the shift of Phe[6.52] causes Phe[5.47] to relocate, creating space for the reorientation of Ile[3.40] and Phe[6.44] in the active conformation. The dashed arrows indicate estimated residue shift during activation.

Phe288 contributes the largest surface area (46.6 Å²) to the complex interface with Compound **1**, and its mutation also has the most significant effect on D1R activation (>100-fold drop in potency). On the other hand, Trp321 is further away from Compound **1** and contributes a smaller surface area toward the ligand (22.1 Å²) (Supplementary Fig. 5c), and its mutation has a more moderate effect (10–20-fold drop in potency). All these data support our proposed activation mode through a network of aromatic residues, which does not require hydrogen bond interactions with the conserved Ser198[5.42] and Ser202[5.46] residues on TM5. To be noted, in our structure, the Trp321[7.43] side chain rotamer will not be compatible with the inactive-state position of Trp285[6.48] estimated by comparison with the respective residue in the β2AR inactive-state structure (Supplementary Fig. 5f)[44]. Since the interaction between Trp321[7.43] and Compound **1** is relatively weak due to their distance, its rotamer in this structure is likely to be a result rather than cause of D1R activation. It is likely to adopt a different rotamer in inactive-state of D1R. Under this proposed activation mode involving a network of aromatic interactions, the efficiency of D1R activation depends on the positioning of the furo-pyridine ring, which is likely to be determined by other parts of the ligand due to its relatively rigid backbone. Indeed, ligands bearing this scaffold can behave like partial and full agonist with very subtle changes towards the distal end of the furo-pyridine ring[6,17].

## Discussion

We report here the crystal structure of D1R in complex with Gs and a non-catechol agonist. The structure shares commonalities with other active-state structures of aminergic receptors, while it also reveals unique features of D1R. The non-catechol agonist Compound **1** binds in an extended conformation, spanning from the orthosteric pocket to ECL2, and the structure reveals the unique features for both of these regions in D1R: the bulky Trp321[7.43] residue in orthosteric site and the bulged conformation of ECL2 related to D1R's unique sequence. The development of subtype selective D1R/D5R orthosteric ligands can benefit from these features.

Our structure also shed light on the mechanism for the activation of D1R by non-catechol agonists. An aromatic element of the agonist, through interaction with two aromatic residues, Phe288[6.51] and Phe289[6.52] on TM6 of D1R, initiates the conformational change of the Ile111[3.40] and Phe281[6.44] of the PIF motif, without the requirement of hydrogen bonding with Ser198[5.42] and Ser202[5.46], or the inward shift of Pro206[5.50] on TM5. A similar conformation of the key aromatic residues has been reported for the α2B adrenergic receptor bound to dexmedetomidine and D2R bound to bromocriptine, both of which are non-catechol agonists that lack hydrogen bond interactions with Ser[5.42] and Ser[5.46] [39,45] (Supplementary Fig. 6). Therefore, this mode of activation can be extrapolated to other aminergic receptors due to their similarities in key residues. Recently, it was reported that phenethylamine, a trace amine metabolite, with the same structure as dopamine with the exception of the two hydroxyl groups, also possess significant potency as a non-selective dopamine receptor agonist[46].

The development efforts towards this non-catechol agonists scaffold have yielded many close analogs showing a spectrum of efficacy from partial agonists to full agonists, and functional selectivity from unbiased to Gs-biased[6,17,42]. Compounds with this scaffold, featured by their rigid extended conformation as

revealed in our structure, offers a good ensemble of ligands to study the structural basis for regulating activation efficacy and biased signaling of the D1R. It was proposed that efficient coupling to G protein and β-arrestin requires distinct conformations of the extracellular pocket in GPCRs, thus a ligand with an extended binding pose is more likely to facilitate or interfere with either one of the conformations, resulting in biased signaling[47,48]. It was also shown that interactions with certain regions on TM6 and TM7 may be critical for regulating biased signaling[49,50]. While in our structure, the relatively small uracil ring of Compound 1 does not make any direct interaction with TM6 or TM7, our docking analysis with reported analogs of Compound 1[6,17] predict that when substituting the uracil ring in Compound 1 for larger ring structures with particular arrangements of heteroatoms, they may form additional polar interactions with Ser310[7.32] and/or Asp314[7.36] on TM7, or may facilitate the interaction between the furo-pyridine ring and Asn292[6.55] on TM6, previously shown to be a critical regulator of partial agonism and biased signaling in β2AR[49] (Supplementary Fig. 7). The distinct interaction pattern for each compound may account for their distinct efficacy profile. Additional structural, biochemical and computational studies with compounds sharing this scaffold may further unlock the mechanism of D1R activation and biased signaling, and facilitate the design of drugs that can modulate D1R function for desired therapeutic outcomes.

GPCRs can be modulated from regions outside their orthosteric binding sites by allosteric modulators. Since allosteric sites are usually less conserved than the orthosteric sites, the targeting of allosteric sites is considered to be a good strategy for developing receptor specific compounds[4,14]. Traditionally, the extracellular vestibule adjacent to the orthosteric site is targeted as the primary allosteric site, as in the case of the M2 muscarinic receptor[51]. In our structure, Compound 1 already occupies much of the extracellular vestibule adjacent to the orthosteric pocket (Supplementary Fig. 8a), which contributes to the selectivity towards D1R/D5R, as discussed above. Recent structural studies revealed that allosteric modulators can bind to multiple extrahelical surface pockets of GPCRs, both extracellular and intracellular, as well as within the lipid bilayer[52–55]. Specifically, the region surrounding ICL2 of D1R is reported to be the binding site for several D1R PAMs, based on mutagenesis and recent structural studies[15,16,56]. In our structure, this extrahelical region on D1R forms a pocket together with the cytoplasmic end of TMs 3 and 4 (Supplementary Fig. 8b). The side chains of Trp123[3.52], Arg130, Lys134, and Met135 form a hydrophobic groove for potential ligand binding. There is a continuous density observed in the groove in the crystal structure, which can be explained by a Polyethylene Glycol 400 molecule in the crystallization environment (Supplementary Fig. 8b). The helical conformation of ICL2 helps to stabilize the D1R active conformation through a hydrogen bond between Tyr131 and Asp120[3.50] in the DRY motif, a conserved interaction among aminergic receptors in their active-state[21]. This helical conformation also facilitates the interaction of Glu132 and Phe129 of D1R-ICL2 with the α5 and αN-β1 hinge region on Gαs, both of which are critical for G protein coupling and activation[47,57–59] (Supplementary Fig. 8c). In the β2AR inactive state, ICL2 is a loop and PAM binding stabilizes the helical conformation of ICL2[60]; therefore, we might expect a PAM binding to this pocket in the D1R would have a similar effect. ICL2 in the inactive or intermediate conformation of other closely related aminergic receptors adopt a loop conformation or become disordered, except when directly involved in crystal packing[35,37,38] (Supplementary Fig. 8d).

## Methods

**Protein engineering for structural studies.** A truncated construct of human dopamine receptor D1R was produced comprising residues 21–347 with T4L fusion at the N-terminus with one alanine residue in between. A leading sequence of β2AR residue 2–20 was further attached to the N-terminus of T4L to increase expression level. A HRV 3C protease cleavage site and linker residues were inserted between the β2AR(2–20) and the T4L. The resulting chimera sequence was subcloned into a modified pFastBac1 vector (Invitrogen), which contained a haemagglutinin (HA) signal sequence, a FLAG tag, followed by three linker residues, before the N terminus of the chimera sequence. The crystallization construct, named D1-105C(−1), has similar affinity towards dopamine and Compound 1 compared to wild-type (Supplementary Fig. 9a).

**D1-Gs heterotrimeric complex formation and purification.** The construct was expressed in *Spodoptera frugiperda* 9 cells in ESF 921 medium (Expression Systems). Cells were infected at a density of $4 \times 10^6$ cells/ml with baculovirus. Culture were grown at 27 °C with constant shaking and harvested by centrifugation 48 h of post-infection.

The harvested cells were lysed in a hypotonic buffer containing 10 mM Tris, pH 8.0, 1 mM EDTA supplemented with 10 μM SCH23390 (Tocris), 2 mg/ml Iodoacetamide and protease inhibitors before use. The membrane was collected and solubilized in buffer containing 30 mM HEPES pH 7.5, 500 mM NaCl, 20% Glycerol, 1% (w/v) n-dodecyl-β-D-maltopyranoside (DDM; Anatrace) and 0.2% (w/v) cholesteryl hemisuccinate (CHS, Steraloids), supplemented with 10 μM SCH23390, 2 mg/ml Iodoacetamide, and protease inhibitors before use. The supernatant containing solubilized D1R was isolated from the cell debris by high-speed centrifugation, and subsequently incubated with FLAG resin overnight at 4 °C in the presence of 2.5 mM CaCl₂. After binding, the resin was washed with 20-column volumes of wash buffer 1 containing 30 mM HEPES pH 7.5, 500 mM NaCl, 0.1% (w/v) DDM, 0.02% (w/v) CHS, and 10 μM SCH23390, followed by another 20-column volumes of wash buffer 1 without SCH23390. The resin was transferred to a tube. Reagents were added in to form the complex, including 100 μM Compound 1 (synthesis quality verified by mass spectrometry and nuclear magnetic resonance spectroscopy, Supplementary Fig 9b), Gs heterotrimeric protein 1 mg/ml (as previously described), nanobody-35 0.2 mg/ml, 10 mM MgCl₂, 1 mM MnCl₂, Lambda phosphatase (NEB) 200 U/ml and Apyrase (NEB) 50 U/ml. The resin and mixture were gently agitated at 4 °C for 3 h and then was packed into column. Washed the column with 10-column volumes wash buffer 1, followed by 12-column volumes of wash buffer 2 containing 30 mM HEPES pH 7.5, 150 mM NaCl, 0.2% (w/v) lauryl maltose neopentyl glycol (LMNG, Anatrace), 0.02% (w/v) CHS. The column was finally washed with 2-column volumes of wash buffer 3 containing 30 mM HEPES pH 7.5, 150 mM NaCl, 0.01% (w/v) LMNG, 0.001% (w/v) CHS. All the wash buffers for the Flag column included 10 μM Compound 1 and 2.5 mM CaCl₂. Protein was eluted with the wash buffer 3 without CaCl₂ and supplemented with 0.2 mg/ml Flag peptide and 5 mM EDTA pH 8.0. Protein was concentrated and applied on Superdex 200 increase column (GE healthcare) with wash buffer 3 supplemented with 100 μM TCEP. Typically, the size exclusion chromatography profile showed a monodispersed peak.

**Lipidic cubic phase crystallization.** Protein samples of the D1R-Gs-Compound 1 complex, at the concentration of 20 mg/ml, were reconstituted into lipidic cubic phase (LCP) by mixing with molten lipid (10% (w/w) cholesterol (Sigma), 90% (w/w) 1-(7Z-tetradecenoyl)-rac-Glycerol (MAG7.7, Avanti Polar Lipids) in a mechanical syringe mixer at a ratio of 1:1 (w/w) protein solution to lipid. LCP crystallization trials were performed using Gryphon crystallization robot (Art Robbins Instruments). The 30 nl bolus was covered with 800 nl of precipitant solution. The 96-well glass sandwich plates were incubated at 20 °C. Initial crystal hits were found from MemStart + MemSys HT-96 screen (Molecular Dimensions) with condition 0.1 M NaCl, 0.1 M MgCl₂, 0.1 M sodium citrate pH 5.5 and 30% (w/v) PEG400. After optimization, crystals grew in 0.1 M MES pH 6.5, 0.1 M ammonium dibasic citrate, 18–25% (w/v) PEG400, 1 mM TCEP, 10 μM Compound 1 and with or without positive modulator in the precipitant solution. Crystals grew 5–7 days to full size (about 20–30 μm across all three dimensions) and were collected using MiTeGen micromounts and flash-frozen in liquid nitrogen for data collection.

**Crystallographic data collection and structure determination.** X-ray data were collected at the 23-ID-D and 23-ID-B beamline at the Advance Photon Source with JBluIce software (ID-D or ID-B version, v2015.1). Microbeam of 10 μm diameter was used to acquire all diffraction data. Owing to radiation damage, only 5–30 degrees of rotation data were collected from each crystal. All data were processed with the XDS package[61]. A 3.8 Å dataset was obtained by merging diffraction data from 15 crystals. The space group was determined to be P2₁22₁. Molecular replacement was performed using the program Phaser[62] in CCP4 package, with separate receptor, T4L, and Gβγ chains from β2AR-Gs structure (PDB ID: 3SN6) as the search model. Iterative rounds of model building were performed using COOT[63], and structural refinement were performed using Refmac5[64] and Phenix[65]. During refinement, secondary structure restraints and a single group of translation/libration/screw (TLS) parameters were applied. The quality of the structure was assessed using Molprobity[66]. Data processing and structure refinement statistics are shown in Supplementary Table 1.

**cAMP assay.** Functional studies were performed using CisBio (Codolet, France) HTRF dynamic-cAMP assay kit in 384-well format with 600–800 cells per well. Cells

were pre-incubated with test compound incubated for 15 min and then agonist (typically dopamine) added, and the incubation continued for a further minute after which cAMP levels were determined in the presence of IBMX (0.1 mM final). Results were always compared to wild type control and data was analyzed in GraphPad PRISM5.0 using 4-paramater logistic function. Initial data where effects were seen were reproduced in a second batch of transiently transfected cells.

For initial screening assays, the ability of the test compound to potentiate cAMP levels in Lmtk-cells expressing human dopamine D1 receptor was determined in the presence and absence of a concentration of dopamine, which gave a 20% response compared to a maximal effect (10 mM dopamine). Cells (20,000 per well) were incubated for 60 min at room temperature. Results were analysed using the 4-paramater logistic equation and expressed as pEC50 and Erel (the maximal effect relative to the maximal response to dopamine over background expressed as a %).

**MD simulations**. The D1R complex crystal structure had several mainchain gaps and missing sidechain atoms. The missing coordinates were modelled using "Completionist", an implementation of the MEDELLER protocol[67] for the purposes of completing existing 3D models. At the heart of the protocol is a membrane-protein-specific version of PyFREAD for fragment-based loop modelling[68], with missing sidechains modelled using SCWRL3[69] and clashes removed using MODELLER[70]. The two missing loop segments of the transmenbrane domain (chain R), which corresponds to ECL2 and ICL3 were filled with fragments of PDBs 1D3H (chain A) and 2QQI (chain A). The structure was then treated using the Protein Preparation Wizard in Maestro to add hydrogens to facilitate the determination of the appropriate protonation states of the ionizable residues[71], and optimization of the hydrogen bond network[72]. To this concern, Asp103[3.32] protonation is considered to be an important feature of active class A GPCR structures, and we selected the protonated form, based on a predicted pKa of 9.03 by the PropKa software embedded in the Acellera Protein Prepare tool[73]. The structure was then prepared for simulation. The Compound **1** ligand was parameterized with the HTMD parameterize version 1.9.7. The CHARMM36 force field was used and the protein was simulated in a pre-equilibrated $80 \times 80$ POPC bilayer[74]. TIP3P water molecules were used to solvate the system together with $Na+$ and $Cl^-$ ions to obtain an ionic strength of 0.15 M. ACEMD was used with the hydrogen mass repartitioning scheme to run the simulations on GPUs with a time-step of 4 fs[75]. The system was first minimized with 500 steps of conjugated gradient, and then equilibrated at 300 K using an NPT ensemble with a Berendsen barostat at 1 atm for 100 ns. During the equilibration heavy atoms of ligand, protein and lipids were constrained by a 1 kcal/mol/Å² spring constant and gradually released. Particle Mesh Ewald was used for handling long-range electrostatic interactions with a cutoff of 9 Å.

Four simulations of 1 μs each were run in parallel using the production protocol implemented in HTMD on a dedicated GPU cluster. Simulations of the D1-ligand complex were carried out in an NVT ensemble for an aggregated time of 4 μs.

Root mean square deviation (RMSD) analysis of the heavy atoms of Compound **1** was performed to check it's stability in the pocket over the entire length of the simulation, for each of the four simulations, using the crystal structure pose as the reference. A water analysis was performed using VMD plugin WATCLUST version 0.1[76] to search for high occupancy water sites within 4 Å from Compound **1** and Asp103. The calculation was performed using default parameters.

The average distances between Compound **1** and key interacting residues, as well as between residues involved in D1R activation, were calculated over the entire length of the simulation, and for each of the four simulations. To assess the stability of the D1 active structure the distance between Ile111[3.40] and Phe281[6.44] of the PIF motif was monitored as well as the TM3–TM6 distance (considering Cα atoms of residues Arg121 and Val270) and the RMSD from inactive-state of TM7 NPXXY motif (using β2AR inactive-state structure, PDB ID: 2RH1 for comparison) (Supplementary Fig. 5a).

**Reporting summary**. Further information on research design is available in the Nature Research Reporting Summary linked to this article.

## Data availability

Data supporting the findings of this manuscript are available from the corresponding author upon reasonable request. A reporting summary for this Article is available as a Supplementary Information file. Structure and data set in this work have been deposited in the Protein Data Bank under accession code PDB 7JOZ [https://doi.org/10.2210/pdb7JOZ/pdb]. Compound 1 is not immediately available upon request. It will have to be resynthesized. Source data are provided with this paper.

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

## Acknowledgements
We thank the staff at Advanced Photon Source GM/CA beamlines 23ID-B and 23ID-D for their support during X-ray crystallography data collection.

## Author contributions
B.K.K., T.S.K., B.S. and F.L. designed research; D.F., B.S., S.L., Z.A.S., A.V., and M.W. performed research; S.L. and Z.A.S. designed modelling experiments; B.S., D.F., I.F., M.L.C., F.L., S.L., S.K., T.C., T.S., Z.A.S., and B.K.K. analyzed data; and B.S., S.L., Z.A.S., and B.K.K. wrote the paper.

## Competing interests
S.L., Z.A.S., S.K., A.V., M.W., and T.C. are employees of UCB Pharma and may own company stock. B.S., D.F., M.L.C., I.F., T.S.K., and B.K.K. are employees of or consultants for ConfometRx. T.S.K. and B.K.K. cofounded ConfometRx.
