## [Peer Review File · Nature Communications]

Reviewers' Comments:

Reviewer #1:

Remarks to the Author:

This manuscript reports the first crystal structure of a Gs-coupled dopamine receptor (D1), though not at a very high resolution, it is a milestone for the corresponding research field and provides critical structure-function insights. The authors emphasized on the structure features that may be unique to non-catechol agonist bound active conformation in their analysis.

If the authors meant to generalize the specific PIF conformation to a more common activation mechanism of non-catechol agonists of aminergic receptors, a systematic comparison of all the relevant structures is necessary. For examples, the author cited alpha2B receptor structure from ref 55 without providing details, while the recent dopamine D2 receptor active structure is also bound with a non-catechol.

The "five simulations of 50 ns each" were used throughout the manuscript. This is a surprisingly small amount of simulations compared to this group's previous work with the Shaw/Dror groups. Using ACEMD, 4fs, and on GPU, it possibly took less than a day to collect 50 ns. These simulations are inadequate to support some of the conclusions, in particular, the stability of the presented pose, which appears to have ambiguity and was proposed to interact with Asp3.32 through a water molecule. Specifically for this pose issue, I could not find any quantitative analysis of the MD simulations.

Minor issues:

I could not find the pharmacological characterizations showing the engineered D1 construct used in crystallography retains WT-like binding affinity and EC50 for compound1 and dopamine.

The predicted pKa values for Asp3.32 should be provided, not just the two programs used. Was it calculated in the presence or absence of the nearby modeled water? How would the pKa responds to the presence or absence of the water?

In Fig. 4b, it was not very obvious to see "no inward movement of Pro2065.50 is observed". Some pairwise distance comparisons among Calpha atoms of 3.40, 5.50 and 6.44 of those three structures would help the argument. Regardless of this movement, if the intracellular portion of TM5 does not show any difference from catechol bound beta2 structure as shown in Figure 1, what does it imply?

The structure-activity relationships described in ref 17 were heavily cited to justify the presented binding mode of compound 1, some details and examples (in the context of supplementary figure 6) should be provided to guide the readers, e.g., it was difficult to follow:

Line169, "It was shown that any substitution other than a single group at the meta position relative to the linking oxygen atom greatly impaired ligand potency 17,...", what is "any substitution other than a single group at the meta position relative to the linking oxygen atom"?

Line 179, "Structure-activity relationship studies showed that any alternatives for this moiety larger than a joint ring structure in any dimension will significantly lower the potency by 100-fold or more 17,..", what is "any alternatives for this moiety larger than a joint ring structure in any dimension"?

Reviewer #2:

Remarks to the Author:

This manuscript reports on the crystal structure of the human dopamine D1 receptor in complex with a non-catechol partial agonist and Gs heterotrimer. The main findings include the identification of the binding mode of the non-catechol ligand, as well as the observation of an extended conformation of TM5 along with a slightly rotated binding pose of Gs as compared to the previously published crystal structure of b2AR-Gs complex. Based on the structure analysis and limited molecular dynamics simulations, the authors proposed a mechanism of D1R activation by the non-catechol agonist. These results add an additional variant to the repertoire of GPCR-Gs

complexes and can be further leveraged for designing more selective non-catechol agonists for D1R. The major criticism is the low resolution of the structure, which will limit its usability for structure-based drug design, and the lack of functional confirmation of the ligand-binding pose and the proposed mechanism.

Major points

1. Crystallization is prone to artifacts, and low resolution can lead to large errors in the model coordinates. The authors should validate the ligand binding pose and the proposed mechanism by introducing site-specific mutations and checking them with radioligand binding and/or functional assays. For example, it would be important to check how mutations of Phe288, Phe289 and Trp285, speculated by the authors to act as trigger points of activation, affect ligand binding and signaling. Other important residues to check by mutagenesis are Trp321 and Asp103.

2. Molecular dynamics simulations were run for 50 ns, which is extremely short and not very useful. The current standard in the field is to conduct multiple > 1 μ s runs.

3. Modeling a water molecule at 3.8 Å resolution is not justified. There are many density blobs at 2.5 σ in the final mFo-DFc maps; most of them are just noise. At this resolution, it is not possible to distinguish between a direct or water-mediated hydrogen bond between the ligand and Asp103. Moreover, Supplemental figure 3b shows that this water molecule makes hydrogen bonds with both Asp103 and Ser107; in the structure model, the distances from the water molecule to Asp103 is 2.28 Å (too short) and to Ser107 is 3.75 Å (too long). For the reference, the distance between Asp103 and O2 atom of the ligand is 3.61 Å (a bit long but could be considered as a hydrogen bond taking into account the coordinate errors at this resolution).

4. The lipid molecule shown in Supplementary figure 5b does not fit well in the density, does not make any polar contacts with the protein and is oriented almost parallel to the lipid bilayer. Modeling lipids at this resolution is difficult and also not justified unless the density is clear and there is additional evidence. This could be a PEG400 molecule, which is more likely considering the location and orientation. Further speculations that this site can be targeted by positive allosteric modulators should be moved to the discussion section, as this is just a pure hypothesis that was not verified by any experimental data.

5. Since Compound 1 was custom synthesized, please provide analysis of its purity by NMR and/or mass spectrometry.

Minor points

6. Please, provide protein concentration used for crystallization and the average crystal size used for data collection.

7. In the methods, the authors mention that Refmac5 was used for the refinement, however, the provided coordinates were clearly refined by Phenix? Please, describe the refinement procedure and the options that were used for the refinement of this low resolution structure. Was any reference model used for additional constraints?

8. The Ramachandran stats and the clash score are in a marginally accepted range and should be improved. The highest resolution shell in Table 1 is a bit too wide for the resolution; 3.8 – 4.0 would be more appropriate. The number of reflections used for the refinement in the provided pdb file (19,954) does not correspond to the value reported in Table 1 (20,110).

Responses to reviewers

Reviewer #1 (Remarks to the Author):

This manuscript reports the first crystal structure of a Gs-coupled dopamine receptor (D1), though not at a very high resolution, it is a milestone for the corresponding research field and provides critical structure-function insights. The authors emphasized on the structure features that may be unique to non-catechol agonist bound active conformation in their analysis.

If the authors meant to generalize the specific PIF conformation to a more common activation mechanism of non-catechol agonists of aminergic receptors, a systematic comparison of all the relevant structures is necessary. For examples, the author cited alpha2B receptor structure from ref 55 without providing details, while the recent dopamine D2 receptor active structure is also bound with a non-catechol.

Response: Our main finding with the structure is that D1R, and other aminergic receptors, can be activated by non-catechol agonists without hydrogen bonding with Ser5.42, Ser5.46 and the inward movement of Pro5.50. Rather, a main driving force for receptor activation is through the rearrangement of a network of aromatic residues. We added a new figure, Supplementary Figure 6 (see below), comparing the respective aromatic residues in α 2B and D2 receptor structure bound with non-catechol agonists show that the proposed mechanism seems to be common among aminergic receptors.

The conformation of the PIF motif, especially the Ile3.40 and Phe6.44, are largely conserved among all aminergic receptor and are determined mostly by the intracellular coupling G protein.

On a separate note, we updated Fig. 2c and Fig. 3b using the D2 active state structure instead of the inactive structure, which is more relevant in comparison. We also updated Fig. 3c to include D2 structure in the comparison.

The “five simulations of 50 ns each” were used throughout the manuscript. This is a surprisingly small amount of simulations compared to this group’s previous work with the Shaw/Dror groups. Using ACEMD, 4fs, and on GPU, it possibly took less than a day to collect 50 ns. These simulations are inadequate to support some of the conclusions, in particular, the stability of the presented pose, which appears to have ambiguity and was proposed to interact with Asp3.32 through a water molecule. Specifically for this pose issue, I could not find any quantitative analysis of the MD simulations.

Response: Four MD simulations of 1 us each have been performed using the entire complex (with G protein) bound to Compound 1. Stability of the compound have been assessed by RMSD showing it remains bound for a total aggregated time of 4 us. In 3 out of 4 simulations it consistently maintains an average RMSD around 0.5 Å with respect to the crystal pose and of 1 Å in the other one simulation. Majority of the larger RMSD for this specific simulation arises from the flip of the uracil ring while it still occupies similar position.

Distance between the oxygen atom in the linker region of Compound 1 and Asp103 has been monitored over the 4 trajectories and it results in an average distance of 4.55 ± 0.8 Å showing no direct interaction. To test if the interaction was bridged by water molecules a water analysis has been also performed using the VMD plugin WATCLUST. The clustering has identified one water site within 4 Å from the oxygen atom of Compound 1 and Asp103. However, the site displays a low water finding probability value (WFP) of average 2.09, meaning the water has only a slightly higher probability to be seen close to Asp103 than in the bulk. Usually WFP values above 6 are associated to high occupancy water sites. In conclusions waters can access the region but are not involved in a stable interaction.

In the revised structure model, we have removed the water molecule and adjusted related text.

Minor issues:

I could not find the pharmacological characterizations showing the engineered D1 construct used in crystallography retains WT-like binding affinity and EC50 for compound1 and dopamine.

Response: The binding of WT D1 and engineered D1 construct with dopamine and Compound 1 was shown in Supplementary Figure 9 and described in text in method section. The crystallization construct showed no significant change of affinity towards dopamine and Compound 1 compared to wild-type.

The predicted pKa values for Asp3.32 should be provided, not just the two programs used. Was it calculated in the presence or absence of the nearby modeled water? How would the pKa responds to the presence or absence of the water?

Response: The MD system has been rebuilt using PropKa version 1.3. The predicted pKa value of Asp103 is 9.03 so the residue has been protonated. Unfortunately, the forcefield used to perform the MD simulations is not a polarizable forcefield so the protonation state chosen at the beginning is maintained throughout the entire simulation.

In Fig. 4b, it was not very obvious to see “no inward movement of Pro2065.50 is observed”. Some pairwise distance comparisons among C α atoms of 3.40, 5.50 and 6.44 of those three structures would help the argument. Regardless of this movement, if the intracellular portion of TM5 does not show any difference from catechol bound beta2 structure as shown in Figure 1, what does it imply?

Response: The distance between C α atoms of Ile3.40 and Pro5.50 in β 2AR-Gs and D1R-Gs structure is 6.9Å and 7.5 Å, respectively. More importantly, the closest distance of atoms between P5.50 and I3.40 in β 2AR-Gs is 3.8Å, while in D1R-Gs structure is 4.9Å. Therefore, there is no direct interaction of P5.50 and I3.40 in the D1R-Gs structure, which indicates that the activation of D1R by Compound1 is not driven by its interaction with TM5 and not required for the active conformation of Ile3.40 and Phe6.44 of the PIF motif, as are the cases with catechol agonists.

The above discussion has been added to the text.

The conformation of the intracellular portion of the receptor, including TM5, was mostly determined by G protein coupling, therefore, it was not surprising that there’s no difference of TM5 between β 2AR-Gs and D1R-Gs structures.

Comparison of the distances between the C α atoms of 3.40, 5.50 and 6.44 in the β 2AR inactive state structure are not the best indicators since all three residues undergo significant conformational change during receptor activation.

The structure-activity relationships described in ref 17 were heavily cited to justify the presented binding mode of compound 1, some details and examples (in the context of supplementary figure 6) should be provided to guide the readers, e.g., it was difficult to follow:

Line169, “It was shown that any substitution other than a single group at the meta position relative to the linking oxygen atom greatly impaired ligand potency 17,...”, what is “any substitution other than a single group at the meta position relative to the linking oxygen atom”?

Response: We have added a new figure, the current Supplementary Figure 4 (see below) to aid the analysis of structure-activity relationship and edited the sentence to “any substitution at a position other than the methyl group on Compound 1 greatly impaired ligand potency”.

Line 179, “Structure-activity relationship studies showed that alternatives for this moiety larger than a joint ring structure in any dimension will significantly lower the potency by 100-fold or more 17,..” what is “any alternatives for this moiety larger than a joint ring structure in any dimension”?

Response: The new Supplementary Figure 4 also covers this part of the analysis. We have edited the sentence to “Structure-activity relationship studies showed that a fused aromatic moiety at this position is required for strong potency in this scaffold.”

Reviewer #2 (Remarks to the Author):

This manuscript reports on the crystal structure of the human dopamine D1 receptor in complex with a non-catechol partial agonist and Gs heterotrimer. The main findings include the identification of the binding mode of the non-catechol ligand, as well as the observation of an extended conformation of TM5 along with a slightly rotated binding pose of Gs as compared to the previously published crystal structure of b2AR-Gs complex. Based on the structure analysis and limited molecular dynamics simulations, the authors proposed a mechanism of D1R activation by the non-catechol agonist. These results add an additional variant to the repertoire of GPCR-Gs complexes and can be further leveraged for designing more selective non-catechol agonists for D1R. The major criticism is the low resolution of the structure, which will limit its usability for structure-based drug design, and the lack of functional confirmation of the ligand-binding pose and the proposed mechanism.

Major points

1. Crystallization is prone to artifacts, and low resolution can lead to large errors in the model coordinates. The authors should validate the ligand binding pose and the proposed mechanism by introducing site-specific mutations and checking them with radioligand binding and/or functional assays. For example, it would be important to check how mutations of Phe288, Phe289 and Trp321, speculated by the authors to act as trigger points of activation, affect ligand binding and signaling. Other important residues to check by mutagenesis are Trp321 and Asp103.

Response: In the 2018 report (citation 6) that first described this scaffold, comprehensive mutagenesis including mutants of Phe288, Phe289, Trp321 and Asp103, plus other residues, were tested with several close analogs of Compound1 (named PF-2334, PF-6142, PF-8871). We have cited these results briefly in the original manuscript and we have now revised the text to provide more support of the ligand binding pose and proposed activation mechanism. These data are shown in the graph below and are discussed further:

Citation:
6. Gray, D. L. *et al.* Impaired β -arrestin recruitment and reduced desensitization by non-catechol agonists of the D1 dopamine receptor. *Nature Communications* 9, doi:10.1038/s41467-017-02776-7 (2018).

Supplementary Table 6: D1R agonist **fold shift** of cAMP functional potency in D1R single-residue mutants relative to wild-type (EC_{50} mutant/ EC_{50} WT).

→ Mutations that affect this scaffold most significantly.
→ Mutations that affect dopamine efficacy much more significantly.

Test Compound	Experiment #	T59A	K81R	D103A	S107A	I117A	S188A	S188I	S188T	L190A	Y194F	S198A	S199A	S202A	V270A	T273A	F288A	F289A	N292A	W318A	W321A	N334A	
Dopamine	1	0.2	1.1	A.L.D.	15.3	25.6	0.3	1.6	1.0	123.2	1.4	15.9	28.7	52.7	1.0	0.3	441.8	304.0	A.L.D.	21.2	9.4	0.9	
	2			A.L.D.			4.5	4.1	5.2	53.0	3.4	32.0	42.2	249.2			A.L.D.	282.5	A.L.D.	16.0	12.7		
	3			A.L.D.			1.4	2.7	4.7	54.2	1.6	20.0	28.4	36.4			91.6	30.5	A.L.D.	14.2	11.8		
	4			A.L.D.			0.3	1.1	0.6	12.4	1.2	19.1	5.6	11.3			A.L.D.	41.8	34.7	6.5	7.3		
	5			A.L.D.				3.3		36.4		140.5		78.3									
	6			A.L.D.				6.2		37.7		13.4		60.0									
PF-2334	1	B.L.D.	1.3	34.0	2.3	8.3	1.0	4.0	1.7	522.7	1.3	6.0	8.0	1.7	1.3	0.7	351.3	67.3	51.0	12.0	12.3	1.3	
	4			31.0			0.4	6.0	2.0	467.0	1.0	8.0	4.0	1.0			522.0	77.0	27.0	18.0	19.0		
	5			35.9				6.7		1180.0		11.5		1.9									
	6			43.9				5.6		498.0		7.2		1.0									
PF-6142	1	1.0	2.7	31.3	2.3	9.8	2.0	6.7	1.7	153.2	1.7	6.0	8.7	1.2	1.7	0.2	141.2	32.3	23.7	16.3	22.7	1.3	
	3			42.6			2.3	10.6	1.3	155.7	0.8	7.0	10.0	1.6			88.7	44.7	30.6	20.4	21.1		
	5			78.1				8.5		225.0		6.4		2.8									
PF-8871	4			12.6			0.6	8.7	1.8	134.4	1.3	3.3	4.3	1.7			146.3	27.9	30.7	12.5	21.4		
	5			34.4				5.8		522.0		5.7		2.6									
	6			39.7				10.8		598.0		8.6		1.7									

A.L.D. = above limit of detection B.L.D. = below limit of detection

Mutations of the aromatic residues Phe288, Phe289, Trp321 resulted in major reduction (>10 -fold) in potency to activate D1R by these compounds. Among them, Phe288 contributes the largest buried surface area (46.6\AA^2) (calculated by PISA) to Compound 1, and its mutation also has the most significant

effect on D1R activation (>100-fold drop in potency). On the other hand, Trp321 is further away from Compound 1 (Supplementary Figure 5c) and contribute a small buried surface area (22.1Å²), and its mutation has a more moderate effect (10 to 20 fold drop in potency).

For the residues that are predicted to be critical for catechol agonists' binding and efficacy, including Asp103, Ser198 and Ser202, their mutations were also reported here. Consistent with our model, the mutation of Ser198 and Ser202 resulted in mild or no reduction in the potency of non-catechol agonists listed above, but major reduction in the potency of dopamine. The mutation of Asp103, an absolutely conserved residue across all aminergic receptors, resulted in total loss-of-function for dopamine, while this mutant can still be activated by non-catechol agonists at reduced potency, indicating it is contributing to the interaction with the ligand but not as critical as with dopamine, consistent with the structure model.

Among all the residues in the binding pocket, the mutation of Leu190 from extracellular loop 2 (ECL2) (Fig. 2d) was shown to cause the most significant decrease in potency among these non-catechol agonist analogs, supporting the structure model in which Leu190 mediate critical interaction with the uracil and furo-pyridine rings at both ends of the extended Compound 1 molecule, contributing a large buried surface area of 73.7Å².

These discussions have been included in the revised manuscript in their respective sections and highlighted.

Regarding Trp285, we made the mutant of Trp285Ala (W285A), and the results from Western blot and binding assays with radio-labeled SCH23390 indicated that the expression and function of the mutated receptor are severely impaired (see the figure below), making subsequent characterization impossible. Trp285 is located at a critical position deep inside the receptor helical bundle, and its mutation likely destroyed the structural integrity of the receptor.

Characterization of D1R-W285A mutant. (a) The anti-FLAG western blot measuring the expression of recombinant D1R-W285A and wild-type (wt). (b) The binding of 2 μ g membrane to 3 H-SCH23390. (c) Saturation binding of W285A and D1-wt to 3 H-SCH23390. No specific binding to 3 H-SCH23390 was detected with W285A mutant.

2. Molecular dynamics simulations were run for 50 ns, which is extremely short and not very useful. The current standard in the field is to conduct multiple > 1 μ s runs.

Response: Four MD simulations of 1 μ s each have been run using the entire complex (with G protein) bound to Compound 1 reaching an aggregate simulation time of 4 μ s.

3. Modeling a water molecule at 3.8 Å resolution is not justified. There are many density blobs at 2.5 sigma in the final mFo-DFc maps; most of them are just noise. At this resolution, it is not possible to distinguish between a direct or water-mediated hydrogen bond between the ligand and Asp103. Moreover, Supplemental figure 3b shows that this water molecule makes hydrogen bonds with both Asp103 and Ser107; in the structure model, the distances from the water molecule to Asp103 is 2.28 Å (too short) and to Ser107 is 3.75 Å (too long). For the reference, the distance between Asp103 and O2 atom of the ligand is 3.61 Å (a bit long but could be considered as a hydrogen bond taking into account the coordinate errors at this resolution).

Response: We have updated the structure model without the water molecule (together with other modifications). we have updated Fig. 2b and removed Supplementary Fig. 3b, and we have adjusted the text accordingly.

Distance between the oxygen atom in the linker region of Compound 1 and Asp103 has been monitored over the 4 trajectories and it results in an average distance of 4.55 ± 0.8 Å showing no direct interaction. To test if the interaction was bridged by water molecules a water analysis has been also performed using the VMD plugin WATCLUST. The clustering has identified one water site within 4 Å from the oxygen atom of Compound 1 and Asp103. However, the site displays a low water finding probability value (WFP) of average 2.09, meaning the water has only a slightly higher probability to be seen close to Asp103 than in the bulk. Usually WFP values above 6 are associated to high occupancy water sites. In conclusions waters can access the region but are not involved in a stable interaction.

4. The lipid molecule shown in Supplementary figure 5b does not fit well in the density, does not make any polar contacts with the protein and is oriented almost parallel to the lipid bilayer. Modeling lipids at this resolution is difficult and also not justified unless the density is clear and there is additional evidence. This could be a PEG400 molecule, which is more likely considering the location and orientation. Further speculations that this site can be targeted by positive allosteric modulators should be moved to the discussion section, as this is just a pure hypothesis that was not verified by any experimental data.

Response: We appreciate and agree with the suggestion regarding the density at intracellular loop 2 region. We have replaced the original lipid molecule to a PEG400 molecule and it fits the density better, which was shown in updated Supplementary Figure 8 (the original Supplementary Figure 5). We have moved the text regarding positive allosteric modulators to discussion section.

5. Since Compound 1 was custom synthesized, please provide analysis of its purity by NMR and/or mass spectrometry.

Response: We have included the NMR and mass spectrometry report from the synthesis in Supplementary figure 9.

Minor points

6. Please, provide protein concentration used for crystallization and the average crystal size used for data collection.

Response: We have included the respective information in the method section. The protein concentration used in crystallization was 20mg/ml, and the average size of the crystals used for data collection was about 20um to 30um across all three dimensions.

7. In the methods, the authors mention that Refmac5 was used for the refinement, however, the provided coordinates were clearly refined by Phenix? Please, describe the refinement procedure and the options that were used for the refinement of this low resolution structure. Was any reference model used for additional constraints?

Response: We apologize for the confusion. The structure was first determined and refined by Refmac5 a few years ago. During the deposition of the structure coordinate to PDB, we used Phenix for finalization

to fulfill the updated requirements from PDB in terms of file format and other information. No reference model was used during refinement.

The description for the refinement has been updated as following:

Iterative rounds of model building were performed using COOT, and structural refinement were performed using Refmac5 and Phenix. During refinement, secondary structure restraints and a single group of translation/libration/screw (TLS) parameters were applied.

8. The Ramachandran stats and the clash score are in a marginally accepted range and should be improved. The highest resolution shell in Table 1 is a bit too wide for the resolution; 3.8 – 4.0 would be more appropriate. The number of reflections used for the refinement in the provided pdb file (19,954) does not correspond to the value reported in Table 1 (20,110).

Response: We have further refined the model and improved the geometry statistics. Revised Supplementary Table 1 included the updated statistics, including Ramachandran favored at 91.9% (vs. original 90.5%), Ramachandran outlier at 0% (vs. original 0.16%), rotamer outlier at 0% (vs. original 0.26%), and clash score at 7.23 (vs. original 8.36). The Rwork/Rfree were also improved slightly, currently at 25.4%/29.7% (vs. original 26.6%/29.8%). There is no qualitative difference in ligand binding poses and conformation of residues discussed in the text, except for the removal of water molecules and replacement of the lipid molecule to a PEG400 molecule, and therefore, we kept most of the original figures.

We have updated the highest resolution shell statistics in Table 1 to cover the range of 3.80Å-3.97 Å.

The original 20,110 reflections stated in Table 1 included the reflections set aside as free R set. We edited the label for the row in Table 1 as “Unique reflections used for refinement” for clarification and put the number as 19,954.

Reviewers' Comments:

Reviewer #1:

Remarks to the Author:

All my comments have been addressed.

Reviewer #2:

Remarks to the Author:

The authors answered all my questions. Just a brief note, since several authors are employed by pharmaceutical companies, I would expect to see a statement of competing interests or a conflict of interests, which I have not found in the manuscript.